

# The effect of landscape on functional connectivity and shell shape in the land snail *Humboldtiana durangoensis*

Benjamín López, Omar Mejía and Gerardo Zúñiga

Laboratorio de Variación Biológica y Evolución, Departamento de Zoología, Escuela Nacional de Ciencias Biológicas, Instituto Politécnico Nacional, Mexico City, Mexico

## ABSTRACT

The populations of *Humboldtiana durangoensis* have experienced a drastic reduction in the effective population size; in addition, the species is threatened by anthropogenic activities. For the aforementioned, landscape genetics will serve as a tool to define the potential evolutionarily significant units (ESU) for this species. To complete our objective, we evaluated the effect of cover vegetation and climate on the functional connectivity of the species from the last glacial maximum (LGM) to the present as well as the effect of climate on shell shape. Partial Mantel tests, distance-based redundance analysis and a Bayesian framework were used to evaluate connectivity. On the other hand, geometric morphometrics, phylogenetic principal component analysis and redundancy analysis were used for the analysis of shell shape. Our results suggest that the suitable areas have been decreasing since the LGM; also, vegetation cover rather than climate has influenced the genetic connectivity among land snail populations, although temperature had a high influence on shell shape in this species. In conclusion, vegetation cover was the main factor that determined the functional connectivity for the land snail; however, local selective pressures led to different phenotypes in shell shape that allowed us to postulate that each one of the previously defined genetic groups must be considered as a different ESU.

## INTRODUCTION

Species dispersal can be affected not only by essential processes (e.g., the movement, mating and reproductive fitness of the individuals) but also by ecological and topographical factors (e.g., abiotic variables, land cover, line features and landforms) associated with the landscape (*Manel et al., 2003*; *McRae et al., 2008*). Especially in land snails, dispersal is a process that is highly dependent on a set of variables associated with the landscape, such as climate and vegetation cover, which represent a high physiological cost for the snail (*Dörge et al., 1999*; *Schweiger, Frenzel & Durka, 2004*; *Hylander et al., 2005*; *Aubry et al., 2006*). Thus, in a heterogeneous landscape, the differentiation between populations may be increased not only by the historical events and microevolutionary factors but also by the ecological and topographical factors that

Corresponding author
Omar Mejía,
homarmejia@hotmail.com

determine the habitat or structural connectivity (*McRae, 2006*; *McRae et al., 2008*; *Bell et al., 2010*).

Due to their low vagility, patchy distribution and preference for particular microhabitats (*Dörge et al., 1999*; *Hylander et al., 2005*; *Aubry et al., 2006*), land snails are excellent models for exploring the effects of landscape on the movement of individuals among suitable patches, or in other words, on the functional connectivity (*Tischendorf & Fahrig, 2000*). The effect of the Pleistocene climate changes on the phylogeographical structure and demographic history of land snails has been widely documented (*Ross, 1999*; *Haase et al., 2003*; *Davison & Chiba, 2006*; *Holland & Cowie, 2007*; *Dépraz et al., 2008*; *Guiller & Madec, 2010*); as well as changes in vegetation cover that have caused a decline in abundance and species density (*Hylander, Nilsson & Göthner, 2004*). However, neither the effect of vegetation cover nor the effect of the climate on functional connectivity have been explored yet.

The snails of the genus *Humboldtiana* represent a group of nearly 60 species that have an insular distribution in the mountainous regions from South Texas and New Mexico to Central Mexico (*Thompson, 2006*; *Mejía & Zuniga, 2007*). Many species have very small ranges, with the exception of three species that are widely distributed (*Mejía, López & Reyes-Gomez, 2018*). *H. durangoensis* is distributed in the Madrense Centro ecoregion of the Sierra Madre Occidental in Durango state, mainly in cold temperate forests in an altitudinal gradient ranging from 1,600 to 2,800 m asl. This vegetation community has historically been exploited in Durango state and has also experienced droughts and fires that have led to fragmentation and habitat loss (*Aragón-Piña et al., 2010*). For these reasons, forest loss has turned into a global conservation issue due to its effect on biodiversity (*Fahrig, 2003*).

Conservation efforts in several countries have traditionally been focused on "surrogate" species, which can create the umbrella effect for other sympatric species and, at the same time, serve to attract attention and funding (*Caro & O' Doherty, 1999*). Illustrative examples of this situation in Mexico are the efforts to recover the tiny vaquita porpoise (*Phocoena sinus*) and the Mexican wolf (*Canis lupus baileyi*). Nevertheless, very few efforts have been conducted to preserve "non-charismatic species" such as land snails. In fact, none of the nearly 1,500 species of native land snails that occur in Mexico (*Thompson & Hulbert, 2011*) are included in the Mexican law for endangered species or in the IUCN Red List, a situation that highly contrasts with European land snails (*Cuttelod, Seddon & Neubert, 2011*); at the same time, few studies of the phylogeographic structure or population genetics have been performed with Mexican land snails (*López, Gómez & Mejía, 2017*; *López, Zúñiga & Mejía, 2019*).

On the other hand, while there is a lack of agreement on how to define an evolutionarily significant unit (ESU) (but see the review in *Fraser & Bernatchez (2001)*), we agree with those proposals that suggest that ESUs must include genetic, ecological and morphological differentiation (*Crandall et al., 2000*) that reflect the adaptive distinctiveness. Previous articles have evaluated the population genetics and phylogeographic structure of *H. durangoensis* in the Madrense Centro region using

microsatellite DNA markers and mitochondrial and nuclear DNA (*López, Gómez & Mejía, 2017*; *López, Zúñiga & Mejía, 2019*). The microsatellite analysis recovered seven genetic groups and signals of a strong genetic bottleneck in the populations, while the mitochondrial and nuclear DNA sequences found three main genetic groups that also showed signals of drastic reduction in the effective population size.

To evaluate the effects of vegetation cover and local climatic variables on the genetic differentiation of the snail *H. durangoensis*, we analyzed the functional connectivity in three temporal frames: the last glacial maximum (21,000 years bp), the middle Holocene (6,000 years bp) and the present. In addition, we evaluated the effect of the climate on shell size and shape using phylogenetic comparative methods. Despite the lack of agreement regarding the effects of the climate on shell traits, a strong relationship between the phenotype, genetic variation and climate would be expected (*Dowle et al., 2015*), because land snails as other groups with low vagility and dispersal abilities, tend to develop local morphological adaptations due to restrictive gene flow (*Fitzpatrick, 2012*; *Pfenninger & Posada, 2002*). Both approaches together will allow us to postulate the ESU for this land snail in the Sierra Madre Occidental in Western Mexico.

## METHODOLOGY

### Resistance surfaces

The geographic centroids of each one of the seven genetic groups of *H. durangoensis* previously defined by microsatellite loci by *López, Gómez & Mejía (2017)* were used to determine the effect of the landscape on functional connectivity (Fig. 1). Whereas the landscape can include a large number of variables, in the present work, we followed two approximations to evaluate the functional connectivity between snail populations. The first was to use an approximation of the Grinnellian niche defined from a set of bioclimatic variables (*Bell et al., 2010*; *Ortego et al., 2012*; *Poelchau & Hamrick, 2012*); the second was to analyze the effect of vegetation cover, because it is known that it affects the dispersion of terrestrial snails (*Labaune & Magnin, 2002*; *Armbruster, Hofer & Baur, 2007*; *Ström, Hylander & Dynesius, 2009*; *Edworthy et al., 2012*; *Kappes et al., 2009*), especially in mountain populations where periods of glaciation and deglaciation promoted the contraction and expansion of vegetation cover (*Armbruster, Hofer & Baur, 2007*). In both cases, the different models were generated for three different time frames, including the current period and two time periods representing the extreme conditions experienced during the late Quaternary: the middle Holocene (6,000 years bp), which was warmer and wetter than the present, and the last glacier maximum (LGM), which was characterized by dry and colder climates (21,000 years bp).

To reduce the error in the parameterization, validation and comparison of the models (*Barve et al., 2011*), the available geographic space for the taxon (M) was defined as the Ecoregion Madrense Centro (*González-Elizondo et al., 2013*). Grinnellian niche models were constructed with the 19 climatic variables available in WorldClim (*Hijmans et al., 2005*) and the 18 climatic and topographic variables available in ENVIREM (*Title & Bemmels, 2018*). The models were made at a resolution of 30 arc-seconds, but in the case of the LGM, the variables were used at their native resolution of 2.5 min, and a bilinear
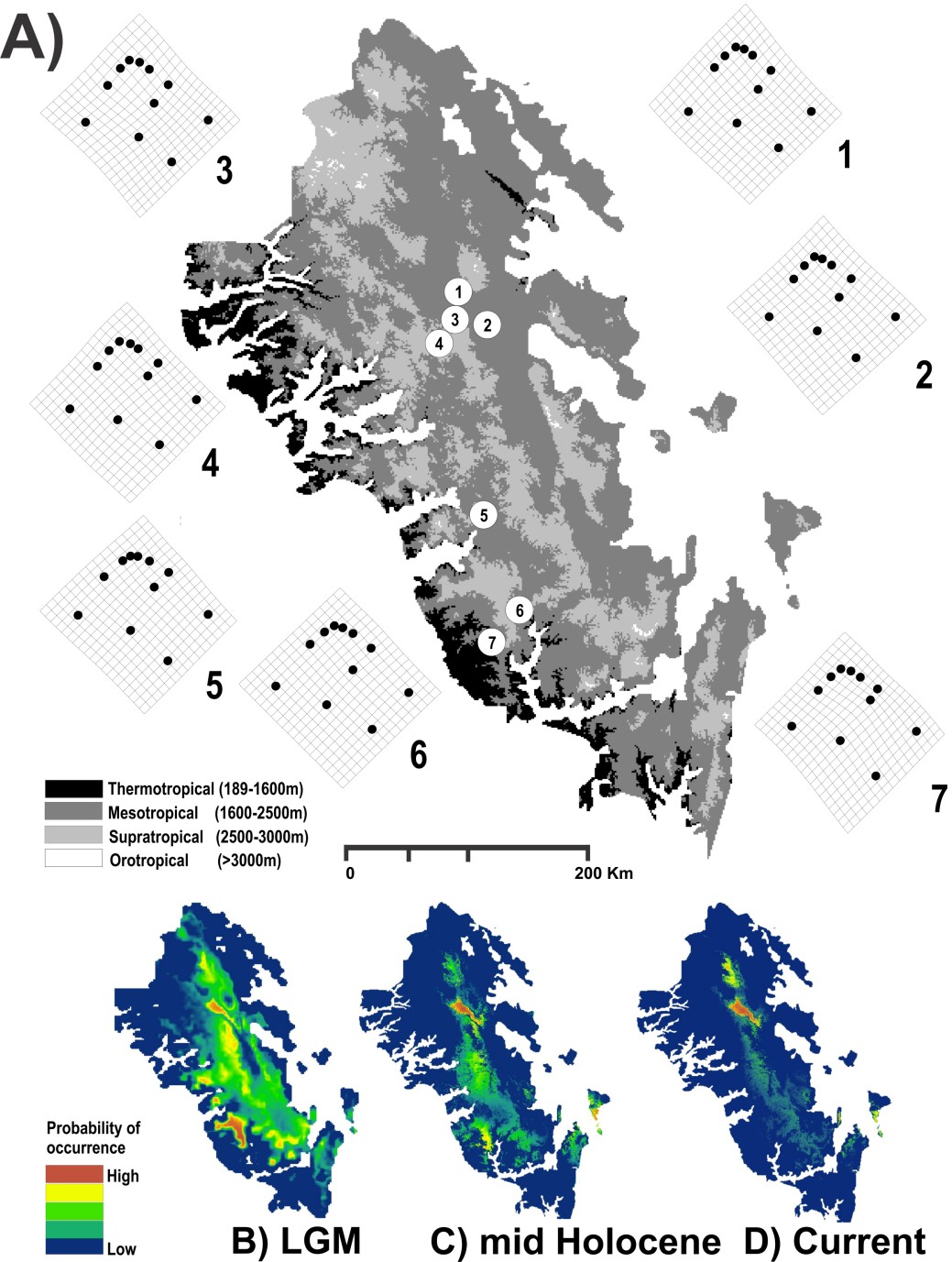

**Figure 1 Study area used.** Geographic map of the Region Madrense Centro in the Mexican state of Durango. (A) A digital elevation model (DEM) was used to highlight the different ombrothermal horizons defined by *Macías-Rodríguez, Giménez de Azcárate-Cornide & Gopar-Merino (2017)*. The circles represent the geographic centroid for each one of the seven genetic groups of *Humboldtiana durangoensis* defined from microsatellite markers in *López, Gómez & Mejía (2017)*: (1) Guanaceví. (2) Los Herreras. (3) Potrero. (4) Topia. (5) Progreso. (6) El Salto and (7) Las Peñas. The deformation grids around the genetic groups represent the average shape of each one of the genetic groups. In the lower section a suitability distribution map from Maxent is showed for the three temporal frames used in this study assuming a minimum training presence from the model (B) Last Glacial Maximum (LGM) (0.172), (C) Mid Holocene (0.369) and (D) Current time (0.347).  

interpolation was performed to decrease the resolution to 30 arc-seconds with the disaggregate function of the raster library ver. 2.6-7 in R (*Hijmans, 2017*). The atmospheric circulation model used was the MPI-ESM-P, since it has shown better performance with respect to other models of circulation (*Tang et al., 2017*). The bioclimatic variables were clipped to the geographic space with the crop and mask functions of the raster library ver. 2.6-7 in R (*Hijmans, 2017*).

## Species niche model

The environmental suitability areas were defined by a maximum entropy algorithm (MAXENT v. 3.2.19, *Phillips, Anderson & Schapire, 2006*) from 28 records of *H. durangoensis* available in museums and our own collections. We selected this algorithm because it produces reliable results even with a small quantity of data (*Elith et al., 2006*; *Heikkinen et al., 2006*; *Hernandez et al., 2006*). In a preliminary analysis, the 19 WorldClim and 18 ENVIREM variables were included with the default parameters and log output to minimize the correlation and maximize their contributions to the model. The relative importance of each variable was determined from its percentage of contribution and for the loss of predictive power when each variable was excluded using a jackknife test. In addition, to select those variables with correlation coefficients lower than 0.6, environmental information was extracted from each geographic point, and a Pearson correlation test was performed with the function correlation test in the psych library of R (*Revelle, 2018*). Thus, the geographic distribution model was obtained with the selected variables and assumed 10,000 pseudoabsence points separated by 1 km from the presence records (*Barbet-Massin et al., 2012*). The statistical evaluation of the model was carried out in 10 repetitions and the data were partitioned into 75% for training and 25% for evaluation with a logistical output. The predictive power of the model was evaluated using a partial ROC test with 100 bootstrap replicates (*Barve, 2008*). Finally, the suitability area available for the species in each temporal frame was estimated with the DEM surface tools in ArcGIS 10.

## Vegetation models

The random forest (RF) classification algorithm was used to obtain the modeled vegetation cover (*Breiman, 2001*). This method categorizes a set of data based on the classification and regression of the trees from a bootstrap analysis (*Breiman, 2001*). The *INE-INEGI (1997)* vegetation cover map was used as an input file. Because this classification contains many vegetation types for the Madrense Centro ecoregion, prior to the analysis, the vegetation types were reclassified into five categories based on the ombrothermal horizons of the Sierra Madre Occidental (*Macías-Rodríguez, Giménez de Azcárate-Cornide & Gopar-Merino, 2017*): (1) temperate forests, (2) cold temperate forests, (3) grasslands, (4) tropical forests and (5) drylands. The model was trained to take into account the variables sets of BIOCLIM and ENVIREM and to select only those that explained more than 50% of the variation based on the mean decrease accuracy criterion; these models were made with the RF 4.6-14 library in R (*Breiman, 2001*).

## Isolation by resistance

To evaluate the resistance of the landscape between the genetic groups, the resistance isolation model (IBR) was implemented in CIRCUITSCAPE 3.4.2 (*McRae, 2006*). This method produces a resistance/conductance matrix between the pairs of sites that are obtained by assigning an arbitrary resistance/conductance value per pixel corresponding to the relative resistance of the landscape to the genetic flow. The result was a resistance value that depended on the distance between the localities, the number of possible pathways and the heterogeneity of the landscape (*McRae, 2006*). The following resistance values were assigned to the forest structure: 60 (cold temperate forests), 110 (temperate forests), 200 (grasslands), 300 (tropical forests) and 360 (drylands).

For the surface derived from the niche modeling, resistance values were assigned considering five symmetrical categories defined by the range between the minimum training presence and the highest suitability value obtained by the Maxent model. The values were assigned with the ifelse and raster R libraries (*Hijmans et al., 2005*), and three different approaches were used to evaluate the relationship between the paired $F_{ST}$ values among the seven genetic groups (*López, Gómez & Mejía, 2017*) and resistance values. Three matrices were considered in this analysis: the genetic paired distances, the Log10-transformed Euclidean geographical distances, and the paired resistance distances obtained from CIRCUITSCAPE for the two evaluated resistance surfaces (climate and vegetation). In the first approximation, the Mantel partial test was used to evaluate the effects of the two variables while controlling for the effect of a third. The significance of the partial correlation of the Mantel test was obtained by 1,000 random permutations using the partial mantel test function of the NFC library (*Bjørnstad, 2013*). In the second approach, a distance-based redundancy analysis (dbRDA) was used in the vegan 2.5 library (*Oksanen et al., 2013*) considering the genetic distances, geographic distances, and the effect of vegetation cover, as well as the effect of the climatic distances on the mean of the resistance values (*Noguerales, Cordero & Ortego, 2016*). The characterization of the environmental space was performed with the random Points function in R that generated 1,000 random geographic points and with the extract function to obtain the climatic point value per site. Then, the main function in R was used to perform the PCA, and later, the dist function in R was used to obtain the eigenvalues of the environmental distances for the first three components considering only the loadings of the geographic points corresponding to the genetic groups; finally, the significance of the dbRDA was evaluated with the anova.cca function in R. Lastly, in the third approach, given that in the two previous analyses the climate component was not significant (see Tables 1 and 2), we only evaluate the effects of geography (G), vegetation cover (E) and both (G + E) on functional connectivity through a Bayesian framework implemented in the SUNDER 0.0.4 library (*Botta et al., 2015*). The algorithm implemented in SUNDER assumed that the covariance of the allelic frequencies among the populations would decrease as a function of the geographical and environmental distances (*Botta et al., 2015*). Thus, to estimate the effect of the set of G, E, and G + E variables, 10 independent chains

**Table 1 Pairwise comparison of Circuitscape.** Pairwise comparison of the resistance values obtained with Circuitscape using the vegetation cover as resistance surface.

| Pair | LGM | Mid Holocene | Current |
|------|------|------|------|
| 1, 2 | 209.87 | 308.20 | 305.42 |
| 1, 3 | 196.76 | 318.14 | 294.33 |
| 1, 4 | 169.83 | 227.74 | 226.22 |
| 1, 5 | 309.86 | 359.06 | 356.54 |
| 1, 6 | 428.33 | 507.67 | 518.36 |
| 1, 7 | 371.76 | 424.61 | 420.29 |
| 2, 3 | 235.52 | 357.59 | 334.44 |
| 2, 4 | 182.15 | 243.87 | 247.67 |
| 2, 5 | 284.65 | 341.65 | 344.16 |
| 2, 6 | 402.27 | 489.54 | 505.29 |
| 2, 7 | 345.54 | 406.29 | 407.04 |
| 3, 4 | 164.42 | 236.34 | 213.96 |
| 3, 5 | 308.73 | 377.72 | 352.35 |
| 3, 6 | 427.46 | 526.63 | 514.47 |
| 3, 7 | 370.94 | 443.65 | 416.48 |
| 4, 5 | 224.20 | 238.48 | 240.16 |
| 4, 6 | 343.37 | 387.74 | 402.65 |
| 4, 7 | 286.93 | 304.84 | 304.74 |
| 5, 6 | 209.67 | 251.50 | 264.47 |
| 5, 7 | 159.03 | 175.71 | 173.63 |
| 6, 7 | 162.45 | 227.27 | 239.57 |

Note:
The numbers in the first column correspond to the geographic centroid of each one of the seven genetic groups recovered by *López, Gómez & Mejía (2017)*: (1) Guanaceví. (2) Los Herreras. (3) Potrero. (4) Topia. (5) Progreso. (6) El Salto and (7) Las Peñas.

**Table 2 Isolation by distance and resistance.** Mantel partial test of the effect of isolation by distance (IBD) and isolation by resistance (IBR) from climate and vegetation surfaces on the genetic differentiation of *Humboldtiana durangoensis* populations for the three temporal frames used in this study.

| Resistance model | Comparison | LGM | | Mid Holocene | | Current | |
|------|------|------|------|------|------|------|------|
| | | R | p | R | p | R | p |
| Climate | fst vs. resistance | 0.474 | 0.094 | 0.368 | 0.122 | 0.418 | 0.111 |
| | fst vs. geogra\|resistance | 0.182 | 0.321 | 0.39 | 0.081 | 0.295 | 0.191 |
| | fst vs. resistance\|geogra | 0.105 | 0.398 | −0.181 | 0.362 | −0.074 | 0.476 |
| Vegetation | fst vs. resistance | 0.057 | 0.382 | −0.144 | 0.29 | −0.111 | 0.344 |
| | fst vs. geogra\|resistance | 0.782 | **0.005** | 0.796 | **0.003** | 0.784 | **0.009** |
| | fst vs. resistance\|geogra | −0.699 | **0.012** | −0.725 | **0.013** | −0.706 | **0.021** |

Notes:
$R$, Spearman correlation coefficient between pairwise genetic distances ($F_{ST}/(1-F_{ST})$) and the Euclidean distance from the geography and pairwise resistance of CIRCUITSCAPE. $p$, Statistical significance obtained from 1,000 replicates. Bold indicates comparisons that were signifcant with $p < 0.05$.

with $10^7$ iterations and sampling every 1,000 steps were used with uniform priors with large upper bounds (*Botta et al., 2015*).

## Shell morphometrics

A total of 129 shells of *H. durangoensis* adults from the seven genetic groups used by *López, Gómez & Mejía (2017)* were analyzed: Las Peñas (20 shells), El Salto (8 shells), Progreso (3 shells), Topia (7 shells), Potrero (46 shells), Los Herreras (25 shells) and Guanaceví (20 shells). The shape of the shell was obtained using two approaches: a classical approach that assumed four linear shell measurements (height, SH; width, SW; aperture height, AH; maximum aperture width, AW) obtained with a digital micrometer with an accuracy of 0.01 mm; in addition, globosity (G = SH/WD), spiral height (SP = SH−AH) and shell volume (V) were calculated (Fig. S1). These variables were Log10 transformed to remove the size effect following the method described by *Mosimann (1970)*. Finally, the eigenvalues of the mean and the centroid values for each one of the genetic groups were recovered from a principal component analysis (PCA) for posterior analysis (*Hartigan & Wong, 1979*). On the other hand, the shell shape was evaluated from 11 landmarks according to *Mumladze, Tarkhnishvili & Murtskhvaladze (2013)* (Fig. S1). Following the method proposed by *Kistner & Dybdahl (2013)*, a total of five photos were taken per individual to eliminate the error associated with the orientation. The *X/Y* coordinates were digitized in TPSDIG ver 2.12 (*Rohlf, 2008*). The average shape per genetic group and the deformation grids were obtained from a generalized Procrustes analysis in order to visualize changes in the shape of the shell with the gpagen function implemented in geomorph 3.0.7 (*Adams, Collyer & Kaliontzopoulou, 2019*). To eliminate the phylogenetic effect on the variation in shell shape, a phylogenetic principal component analysis was performed considering the shell shape of both, classical and geometric morphometrics approaches and a tree based on distances generated from $F_{ST}$ values with the function phyl. pca in phytools (*Revell, 2012*); additionally, the deformation grids of the average shell shape of the genetic groups were ploted in the phylomorphospace with the plotGMPhyloMorphoSpace function implemented in Geomorph. Lastly, to determine whether there was a relationship between the shell shape and environmental conditions, a redundancy analysis (RDA) was performed considering the three matrices generated (means and centroid values for each one of the seven genetic groups as well as the average shape obtained from the geometric morphometrics analysis) with the rda function following the method proposed by *Borcard, Gillet & Legendre (2018)* in the vegan library ver 2.5 (*Oksanen et al., 2013*).

## RESULTS

Six variables made the greatest contribution to the model of the potential distribution: isothermality (Bio3), the minimum temperature of the coldest month (Bio6), the precipitation of the wettest month (Bio 13), the precipitation of the driest month (Bio 14), the precipitation of the coldest month (Bio 19) and the climatic humidity index. For the potential vegetation model, 11 variables were selected: isothermality (Bio 3), temperature seasonality (Bio 4), the annual temperature range (Bio 7), the annual

precipitation (Bio 12), the driest month precipitation (Bio 13), the seasonality of precipitation (Bio 15), the coldest quartile precipitation (Bio 19), the average monthly evapotranspiration potential of driest quarter (PETDriestQuarter), the monthly variability in evapotranspiration potential (PETseasonality), the average monthly evapotranspiration potential of the warmest quarter (PETWarmestQuarter) and the average evapotranspiration potential of the wettest quarter (PETWettestQuarter).

## Environmental suitability and vegetation models

The results obtained for the modeling of the distribution area of *H. durangoensis* in the Madrense Centro region showed that the models constructed for the three temporal frames were satisfactory ($P = 0$). In general, our findings suggested that the areas of environmental suitability had decreased considerably in the last 21,000 years (38,197 km$^2$ or 28.5% of the total area in the LGM, 32,945 km$^2$ or 24.5% in the mid Holocene and 23,620 km$^2$ or 17.6% in the current). Our findings show that at present, the areas with high probability of occurrence are restricted to the northern portion of the distribution area (Fig. 1). The model of vegetation cover generated from the current vegetation map with RF showed that the estimated success rate was 76.77% for the LGM, 77.48% for the middle Holocene and 75.57% for the current period (Table S1). In addition, in the last 21,000 years, a variation in the coverage area of each plant community was estimated, and the temperate forests increased the most, while the grasslands decreased the most (Table S1; Fig. S2).

## Resistance and functional connectivity

The maps generated by CIRCUITSCAPE considering the structure of the vegetation cover suggested that the connectivity routes between the *H. durangoensis* genetic groups in the Central Madrense region have changed little in the last 21,000 years, although in the actual period, the areas of high resistance are larger compared to those in the LGM (Table 1; Fig. S2). The resistance surface from the environmental suitability models for the Mantel test and Mantel partial tests were not significant (Table 2; Fig. S3). On the other hand, when considering the effects of vegetation cover, the Mantel test between the values of $F_{ST}$ and vegetation cover was once again not significant in any of the three time frames; however, the Mantel partial tests yielded significant correlations when controlling for the effects of geography and vegetation cover in the three time periods (Table 2). In the case of the RDA, the marginal tests for the three time frames showed a significant association between the genetic differentiation and geographic distance, explaining 24.26% of the variance, but were not significant when the resistance distances generated from the vegetation cover or from the climatic variables were considered (Table 3). In contrast, in the conditional tests as in the Mantel partial test, a relationship was again observed with the structure of the vegetation cover but not with that of the climate (Table 3). With respect to the results generated by SUNDER, when the climatic component was no longer considered, it was observed that during the LGM, it was the geographic component that best explained the variation, while for the middle Holocene

**Table 3 Distance based redundancy analysis.** Effect of the geographic distance (IBD), vegetation and climate on the genetic differentiation among the seven genetic populations of *Humboldtiana durangoensis* obtained from the distance based redundancy analysis (dbRDA) for the three temporal frames used in this study.

| Marginal tests | | | | Conditional tests | | |
|---|---|---|---|---|---|---|
| **Variable** | *F* | *p* | **% var** | *F* | *p* | **% var** |
| LGM | | | | | | |
| Geographic | 6.086 | **0.02** | 24.26 | | | 0 |
| Vegetation | 0.605 | 0.447 | 3.087 | 14.521 | **0.002** | 33.819 |
| PCA1 | 1 | 0.327 | 4.999 | 0.244 | 0.623 | 1.013 |
| PCA2 | 2.979 | **0.1** | 13.552 | 1.308 | 0.266 | 5.131 |
| PCA3 | 0.098 | 0.748 | 0.515 | 1.536 | 0.238 | 5.956 |
| mid Holocene | | | | | | |
| Geographic | 6.086 | **0.026** | 24.26 | | | 0 |
| Vegetation | 0.404 | 0.537 | 2.083 | 19.986 | **0** | 39.849 |
| PCA1 | 2.106 | 0.166 | 9.976 | 1.916 | **0.177** | 7.286 |
| PCA2 | 0.419 | 0.516 | 2.158 | 0.074 | 0.785 | 0.312 |
| PCA3 | 1.211 | 0.284 | 5.992 | 2.888 | 0.105 | 10.473 |
| Current | | | | | | |
| Geographic | 6.086 | **0.025** | 24.26 | | | |
| Vegetation | 0.235 | 0.636 | 1.224 | 17.844 | **6.00E−04** | 37.704 |
| PCA1 | 0.408 | 0.525 | 2.101 | 0.138 | 0.704 | 0.578 |
| PCA2 | 2.095 | 0.161 | 9.933 | 1.723 | 0.213 | 6.617 |
| PCA3 | 1.48 | 0.231 | 7.228 | 2.82 | 0.108 | 10.258 |

**Notes:**
In the marginal test the effect of each one of the variables was evaluated separately, meanwhile, in the conditional test, the effect of the geographic distance was included as a covariate. *F* represent the proportion of variance, *p* the statistical significance and % var the percentage of variance explained from each variable.
Bold indicates comparisons that were signifcant with *p* < 0.05.

and the actual period, both the geographic component and the vegetation cover were important (Table 4).

## Variation in shell size and shape

The values estimated from the morphometrics classical approach allow us to establish that the populations located in the north of the distribution area (Topia, Potrero, Los Herreras and Guanaceví) had larger sizes and higher spires in comparison with the populations in the center (Progreso) and south (Las Peñas and El Salto) of the distribution area (Table 5). The percentage of variance explained by the first three phylogenetic components was 99.47% for the means of the linear variables, 99.98% for the centroid size and 91.84% for the average shape obtained from the analysis of the geometric morphometrics. Finally, the RDA obtained from the analysis of the first three phylogenetic components was statistically significant ($P < 0.05$). The bioclimatic variables associated with each dataset were different, although in all cases, they were exclusively temperature variables, with the temperature annual range (Bio 7) being the only common variable (Fig. 2). Although it was difficult to establish a pattern, the data retrieved from geometric

**Table 4 Results of the Bayesian inference and model selection obtained from SUNDER to evaluate the relative effect of geography and vegetation cover on the genetic differentiation of the seven genetic groups of *Humboldiana durangoensis*.**

| Period | Iteration | G | | E | | G + E | | |
|--------|-----------|---|---|---|---|-------|---|---|
| | | Likelihood | Bg | Likelihood | Be | Likelihood | Bg | Be |
| LGM | (6, 3, 1) | −8,975.22 | 4.13 | −9,053.74 | 521.65 | −9,044.04 | 4.15 | 1,102.35 |
| mid Holocene | (3, 3, 6) | −6,712.19 | 3.29 | −6,672.19 | 524.79 | −6,638.51 | 3.52 | 2,040.62 |
| Current | (3, 2, 5) | −9,942 | 3.63 | −9,964.55 | 530.97 | −9,890.39 | 3.29 | 1,711.27 |

Note:
G, Euclidean geographic distances; E, Resistance values obtained for the vegetation cover; G+ E, combined effect of both variables. The numbers inside brackets in the iteration column indicate the number of times that each one of the three models has obtained the lower value of likelihood in 10 independent runs. The parameter β represents the magnitude of the effect of the variable on the genetic covariance (lower values indicate a more important effect).

**Table 5 Shell measurements performed.** Average size (in mm) for the four measurements used in this study to evaluate the shell shape of seven genetic groups of *Humboldtiana durangoensis*. Shell height (SH), Shell width (SH) Aperture height (AH), Maximum Aperture width (AW). Additionally, Globosity index (G), Spire Height (SP) and Shell Volume are shown.

| Group | N | SH | SW | ALH | AW | G | SP | V |
|-------|---|-----|-----|------|-----|-----|------|-----|
| Guanacevi | 20 | 32.15 | 34.45 | 21.66 | 19.52 | 0.93 | 10.49 | 3.42 |
| Los Herreras | 25 | 31.94 | 33 | 22.06 | 19.09 | 0.97 | 9.88 | 3.29 |
| Potrero | 46 | 32.86 | 35.4 | 22.32 | 20.34 | 0.93 | 10.54 | 3.47 |
| Topia | 7 | 31.13 | 32.56 | 23.18 | 18.98 | 0.96 | 7.95 | 3.22 |
| Progreso | 3 | 24.11 | 26.16 | 19.11 | 15.81 | 0.92 | 5 | 2.62 |
| El Salto | 8 | 28.94 | 31.3 | 21.28 | 17.97 | 0.93 | 7.66 | 3.16 |
| Las Peñas | 20 | 25.51 | 28.14 | 19.32 | 16.29 | 0.91 | 6.18 | 2.87 |

morphometric analysis allowed us to suggest that larger shells with higher spirals are related to the max temperature of the warmest month (Bio 5), while smaller shells with the lower spirals were related to the temperature annual range (Bio 7) and mean temperature of the wettest quarter (Bio 8) (Fig. 2). Additional support for the aforementioned results proceed from the analysis of the deformation grids in the phylomorphospace, where the populations located to the North (Guanaceví, Los Herreras, Topia and Potrero) tend to have higher spires and higher values of whorl expansion ratio that lead to squared shells, in contrast, the lower spires and the lower values of whorl expansion ratio in the Center (Progreso) and South (El Salto and Las Peñas) populations lead to wider an more rounded shells (Fig 3).

# DISCUSSION

## Effects of the landscape on functional connectivity

The functional connectivity in terrestrial snails was determined by the availability of microhabitats suitable for dispersal. Our findings showed that the variables related to the humidity and relative aridity of the terrain, as well as the precipitation of the driest and the wettest month, had a greater contribution to the potential distribution model

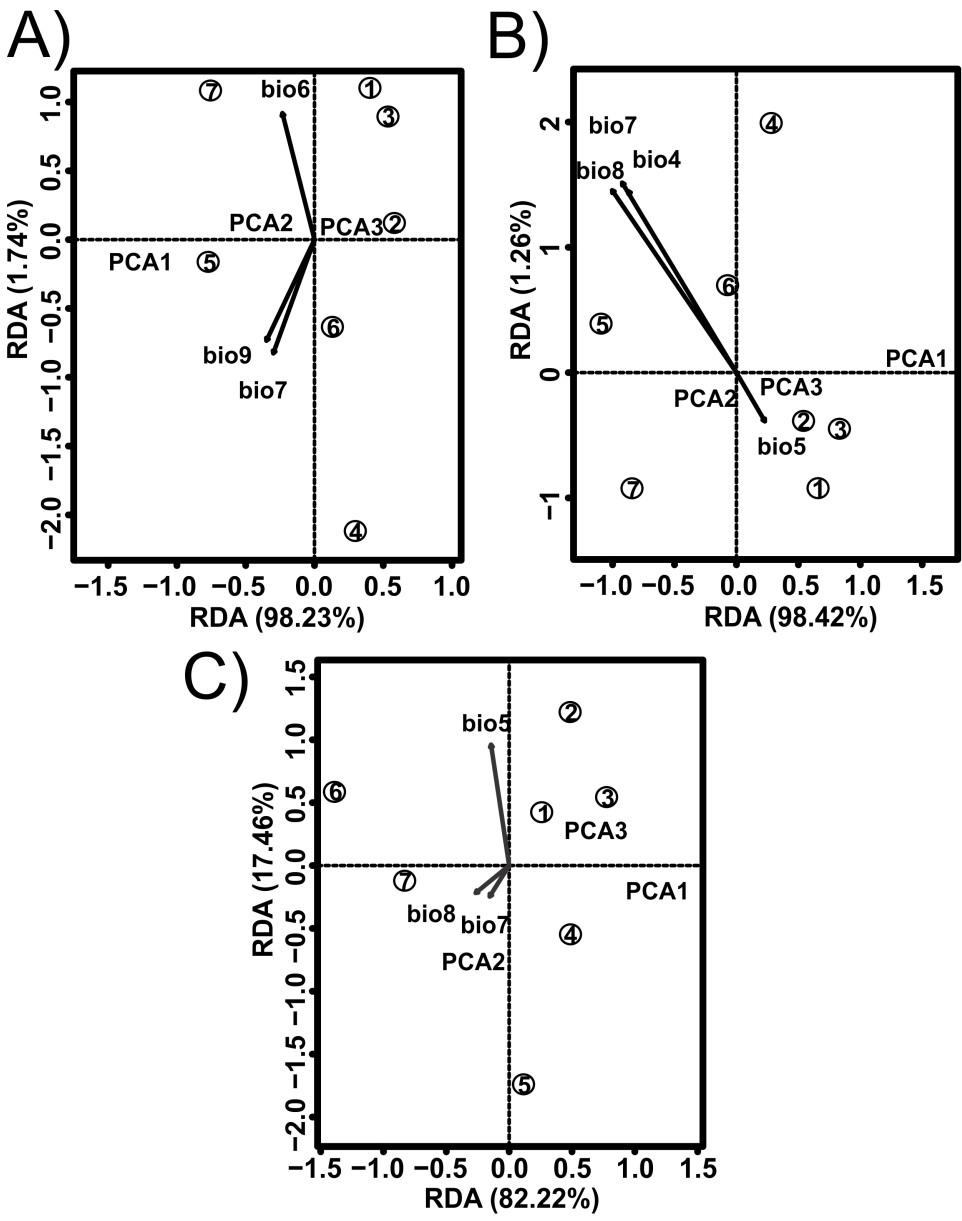

**Figure 2 Redundancy analysis (RDA) for the shell shape of *Humboldtiana durangoensis* between:**
**(A) Average size from traditional morphometrics. (B) Centroid from traditional morphometrics.**
**(C) Consensus shape from geometric morphometrics and climate variables from Worldclim.**
The direction and size of the arrows indicate the correlation between climate variables and RDA axes.
The circles represent the geographic centroid for each one of the seven genetic groups: (1) Guanaceví.
(2) Los Herreras. (3) Potrero. (4)Topia. (5) Progreso. (6) El Salto and (7) Las Peñas.

generated by Maxent. These variables were related to the apparent rupture of the estivation
period in May and to the period of activity and dispersion between July and September, as
has been suggested for other members of the group (*Baur, 1986*; *Aubry et al., 2006*).
However, the climate component defined through the environmental suitability analysis
with the MAXENT maximum entropy algorithm and by the method proposed by

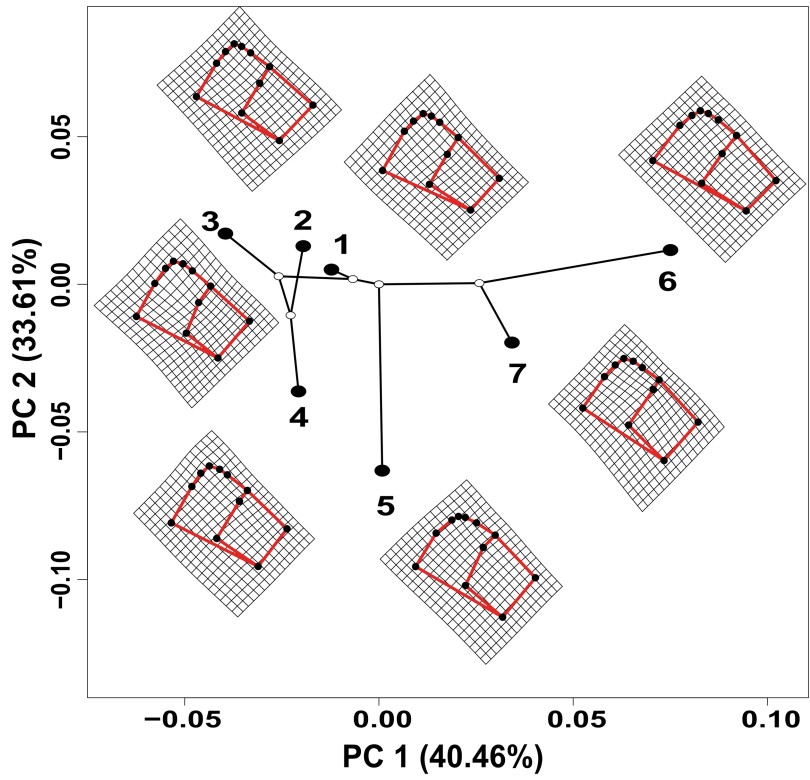

**Figure 3 Deformation grids of the consensus shape of the shells of *H. durangoensis* plotted into the phylomorphospace.** The black circles correspond to each one of the seven genetic groups analyzed. (1) Guananceví. (2) Los Herreras. (3) Potrero. (4) Topia. (5) Progreso. (6) El Salto and (7) Las Peñas. The first two components of the phylomorphospace explained 74.07% of the variation in shell shape.

*Noguerales, Cordero & Ortego (2016)* did not contribute significantly to explaining the functional connectivity of *H. durangoensis* populations.

A possible explanation for this phenomenon might be related to the spatial resolution provided by the bioclimatic layers. It has been demonstrated that the geographic patterns of the areas of environmental suitability in the terrestrial mollusks were particularly dependent on the resolution of the grid, since this increases or diminishes the heterogeneity of the geographic space (*Kadmon & Heller, 1998*). However, the models generated for land snails at a resolution of 30 arc-seconds (1 km²), as used in this study, have been shown to be efficient in explaining the historical demographic reductions that are the consequence of contractions in the areas of environmental suitability (*Horsák et al., 2010*; *Pfenninger et al., 2014*; *Mumladze, 2014*; *Patrão et al., 2015*). In this sense, the areas of environmental suitability for *H. durangoensis* have decreased from 28.5% in the LGM to 17.6% at the present, a result congruent with the population reductions recovered for this species with microsatellite markers and DNA sequences (*López, Gómez & Mejía, 2017*; *López, Zúñiga & Mejía, 2019*). Therefore, although the climate component apparently did not make a significant contribution to functional connectivity, its influence on the taxon cannot be denied because *H. durangoensis* likely

experienced environmental tracking as a consequence of climate change, as has been demonstrated in alpine populations of *Arianta arbustorum* (*Baur & Baur, 2013*).

On the other hand, the RF algorithm has been shown to perform well in predicting the current vegetation types in heterogeneous geographic areas, as it was very robust in relation to the number of classes in which plant communities were clustered, as has been verified by paleopalinological records for models generated for LGM (*Waske & Braun, 2009*; *Rodriguez-Galiano et al., 2012*; *Vanselow & Samimi, 2014*; *Hais et al., 2015*). Thus, the results of efficiency in the assignment to plant categories with the RF algorithm (Table S1) fall within the values obtained in other works (*Waske & Braun, 2009*; *Hais et al., 2015*), suggesting that predictions of vegetation cover in this study are correct. Although our paleovegetation maps apparently did not show significant changes in vegetation cover (Fig. S2), the resistance results from Circuitscape suggested that these changes have occurred and that resistance values have increased from the LGM to the present (Table 1). One of the main limitations of analyses based on resistance surfaces is that the values assigned to each of the categories are arbitrary; however, it has been shown that the assigned resistance values have no effect on the habitat categories in a fragmented landscape (*Schweiger, Frenzel & Durka, 2004*; *Wang et al., 2008*). Consequently, as has been reported for other mountain snails (*Scheel & Hausdorf, 2012*; *Hugall et al., 2002*; *Sherpa et al., 2018*), the altitudinal displacement of plant communities in mountainous regions during Quaternary climate changes could explain the dynamics of functional connectivity in *H. durangoensis* as has been postulated for other species distributed in the SMOc (*Metcalfe et al., 2000*; *Anducho-Reyes et al., 2008*; *Bryson et al., 2011*; *López-González, Correa-Ramírez & García-Mendoza, 2014*).

Based on these findings, we hypothesized that the functional connectivity of *H. durangoensis* on different temporal scales has been promoted by the presence of both temperate and cold temperate forests and that two patterns can be distinguished as has been suggested in *Helix aspersa* and *Cepaea nemoralis* (*Arnaud, 2003*; *Schweiger, Frenzel & Durka, 2004*; *Barahona-Segovia et al., 2019*). The first is a model of isolation by distance on a larger geographic scale (*Pfenninger & Posada, 2002*; *Arnaud, 2003*; *Schweiger, Frenzel & Durka, 2004*), and the second is possible dynamic metapopulation promoted both by environmental and landscape heterogeneity on a fine geographic scale, as has been documented for other land snails (*Arnaud et al., 2001*; *Baur & Baur, 2013*).

## Variation of the shell in *H. durangoensis*

The relationship between shell size and shape in land snails with climatic variables of temperature and precipitation has been widely studied and is well known (see review in *Goodfriend (1986)*). However, while the effect of the genetic component on shell shape variation has been studied (*Goodacre, 2001*; *Dowle et al., 2015*; *Sherpa et al., 2018*), few studies have attempted to control this effect (*Webster, Van Dooren & Schilthuizen, 2012*; *Kotsakiozi et al., 2013*), and none so far have evaluated this effect at the intraspecific level. Our findings showed, after controlling for the genetic effects, that the shell size and shape were determined by climatic variables of temperature and precipitation (Fig. 2).
However, whereas these variables were not significant to explain the genetic relationships among the groups, they suggested that both the phenotype and genotype were the results of independent processes (*Haase & Misof, 2009*); that is, the microhabitat conditions had a great effect on the shell despite the existence of gene flow (*Chiba & Davison, 2007*; *Fiorentino et al., 2013*; *Stankowski, 2013*; *Proćków, Kuźnik-Kowalska & Mackiewicz, 2017*). Thus, whereas it has been suggested that the use of comparative phylogenetic methods at intrapopulation levels may generate poor informative results (*Niewiarowski, Angilletta & Leaché, 2004*), the power of resolution of these methods may depend on the taxon and the assessed trait (*Martins & Housworth, 2002*), as has been found in this study.

In addition, our results suggested that populations with larger shells and apertures are distributed to the north, while populations with smaller shells and apertures were distributed to the south. The altitudinal interval of the sampled localities in the northern region (1,702–2,400 m asl) was lower than the altitudinal interval in which the populations in the southern region were collected (2,587–2,759 m asl), which was consistent with the results previously found in intrapopulation studies of the species of the genera *Arianta*, *Vestia* and *Trochulus* (*Buria & Stahel, 1983*; *Baur & Raboud, 1988*; *Sulikowska-Drozd, 2001*; *Proćków, Kuźnik-Kowalska & Mackiewicz, 2017*), where the populations from colder climates had smaller shells. This could be related to a greater probability of survival of organisms with small shells in unfavorable climatic conditions (*Baur et al., 2014*) and the greater resistance to crystallization temperatures (*Ansart et al., 2014*). At the same time, at higher altitudes, the duration of individual growth time is shorter (*Anderson, Weaver & Guralnick, 2007*; *Proćków, Kuźnik-Kowalska & Mackiewicz, 2017*). However, there were also differences in the sizes of the aperture and the heights of the spires between the north and south regions. These shells attributes could reflect microclimatic conditions, where small apertures tended to occur in the drier and higher altitude regions, meanwhile large apertures and higher spires occured at lower altitudes as has been reported in other species (*Anderson, Weaver & Guralnick, 2007*; *Haase & Misof, 2009*; *Dowle et al., 2015*).

## How many ESUs?

In the literature, only two published works that addressed the definition of the ESUs of land snails have been published (*Holland & Hadfield, 2002*; *Ursenbacher et al., 2010*); however, they did not remove the phylogenetic effects, which impacted their results. In the first study, a fragment of the mtCOI DNA was used and only the phylogenetic trees, genetic distances and AMOVA analysis in the 12 populations of the tree snail *Achatinella mustelina* were recognized as six ESUs that were reproductively isolated and distributed throughout a longitudinal transect of 24 km (*Holland & Hadfield, 2002*). In the second study, which used microsatellite loci and performed a genetic structure analysis, two main clusters were found in *Trochulus aureatus*, although the authors decided to define each one of the nine sampled populations as different ESUs, even though they were separated by less than 200 m (*Ursenbacher et al., 2010*). In opposition, our results suggest that each of the seven genetic groups previously identified by the analysis performed by *López, Gómez & Mejía (2017)* must be considered an ESU, not only because of their

genetic distinctiveness but also due to the phenotypical differences. The removal of the phylogenetic effect shows that temperature and precipitation variables were strong determinants of the shell size and shape of the species, which explained the morphological differentiation (Fig. 2).

## CONCLUSIONS

The main conclusion of this work is that vegetation cover has a high impact on the functional connectivity of the land snail, as does climate, which is a strong determinant of shell shape in this species. Previous studies have found that young restored forests can achieve even higher snail diversities than old unperturbed forests (*Hylander, Nilsson & Göthner, 2004*; *Ström, Hylander & Dynesius, 2009*), although this could depend on survival in microrefugia or dispersal from other patches. Forestry is one of the main economic activities in the state of Durango, Mexico, that exerts strong pressure on the populations of the land snail *H. durangoensis* due to habitat loss and degradation. Nevertheless, the development of comprehensive management plans for the state (*CONAFOR, 2006*) could guarantee the long-term survival of *H. durangoensis*, although further studies need to be performed to evaluate the potential effects of global climate warming on the species.

## ACKNOWLEDGEMENTS

We also thank Angus Davison and two anonymous reviewers for their comments.

### Funding

This work was supported by CONACYT project number 165990. The funders had no role in study design, data collection and analysis, decision to publish, or preparation of the manuscript.

### Grant Disclosures

The following grant information was disclosed by the authors:
CONACYT: 165990.

### Competing Interests

The authors declare that they have no competing interests.

### Author Contributions

- Benjamín López conceived and designed the experiments, performed the experiments, analyzed the data, prepared figures and/or tables, authored or reviewed drafts of the paper, and approved the final draft.
- Omar Mejía conceived and designed the experiments, performed the experiments, analyzed the data, prepared figures and/or tables, authored or reviewed drafts of the paper, and approved the final draft.
- Gerardo Zúñiga conceived and designed the experiments, authored or reviewed drafts of the paper, and approved the final draft.

## Data Availability

The raw data and the code used in this study are available in the Supplemental Files.

## Supplemental Information

Supplemental information for this article can be found online at http://dx.doi.org/10.7717/peerj.9177#supplemental-information.

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
