# Peer review of "The effect of landscape on functional connectivity and shell shape in the land snail Humboldtiana durangoensis"

_PeerJ, doi:10.7717/peerj.9177_

## Round 0.1 · original submission · Minor Revisions

I thought that this paper was thorough and well done. I enjoyed reading it and think it will be a valuable addition to the literature. I have attached several changes I noticed that should be made throughout the manuscript in addition to the suggestions that were proposed by the reviewers.

Additionally, one of the reviewers would like to share code with you per their review. Please let me know if you would like it and I will share it with you.

Let me know if you have any additional questions about the process at PeerJ.

Reviewer 1 ·

Basic reporting

all comments on the pdf file

Experimental design

all comments on the pdf file

Validity of the findings

all comments on the pdf file

Additional comments

Great study

I have a morphometric code in R that I can share if you desire. I mention in my review the use of R for this analysis. Please contact the journal if you are interested in accessing the R code.

I think the manuscript is in a very good shape except for some further analysis in the morphometric side: adding a procrustes analysis and adding deformation grids with the change in shape to figure 1.

Annotated reviews are not available for download in order to protect the identity of reviewers who chose to remain anonymous.

·

Basic reporting

This is a very interesting study where authors evaluate the effect of climate and vegetation cover on the functional connectivity of populations of Humboldtiana durangoensis, a land snail from Mexico, based on information of a previous study in which evidence of population structure was found for the species. Also, the effect of climate in shell size and shape was assessed aimed to distinguish conservation units (ESUs). Although there is no general consensus on how to define ESUs, the authors use herein an interesting approach. The manuscript is clearly written and is in good shape, figures are relevant, and raw data are available as supplemental information.

Experimental design

The research objective is clearly defined, and methods are technically sound and appropriately detailed.

Validity of the findings

Authors evaluated the functional connectivity for the species and provide evidence suggesting that vegetation rather than climate has influenced the availability of suitable habitats, which have decreased since the Pleistocene to present, hypothesizing that functional connectivity has been promoted by the presence of temperate and cold forests. They also found evidence of a relationship among shell size and shape with climatic variables, explained by the latitude and altitude where the populations examined occur. Authors also suggest the seven genetic groups identified in a previous paper should be considered ESUs based not only on the genetic component, but also on the phenotypical differences. Overall, this kind of information is very welcome for the molluscan fauna from Mexico for which conservation efforts are scarce.

Additional comments

- Some references listed in the main text are not in the reference list and vice versa.
- One of the main advantages of geometric morphometrics (GM) is to describe the variation in shape, excluding the size component. In this paper when using GM, shape variation is only described in relation to “size” (e.g. larger shells, higher spirals). In my opinion it would better to describe shape variation in terms of shape (e.g. conical, rounded, squared…).
- I have introduced minor comments and suggestions directly into the attached MS.

---

## Round 0.2 · accepted · Accept

I really enjoyed this paper and think that it will be a valuable addition to the literature. We appreciate your submission. Please let me know if you have any additional questions going forward.

Reviewer 1 ·

Basic reporting

I think the authors made great changes to the article. I loved how they added the deformation grids to their figures. I still think figure S1 should be a figure in the main manuscript and not in the supplementary materials: it's the study system, it deserves a place as a figure in the main manuscript, it makes it easier to understand the deformation grids and the morphometric analysis, it's a beautiful picture. I think figure S1 should be figure 1.

Experimental design

The comments on my first review were followed by the authors and they did a great job

Validity of the findings

As I mentioned the first time, I think the findings are clear and important.

·

Basic reporting

The authors have revised the manuscript and addressed all reviewers’ suggestions. I have no additional comments, and I think the manuscript should be accepted. Congratulations to the authors for their excellent work.

Experimental design

No additional comments.

Validity of the findings

No additional comments.

Additional comments

No additional comments